# Mechanism and Role of Endoplasmic Reticulum Stress in Osteosarcoma

**DOI:** 10.3390/biom12121882

**Published:** 2022-12-15

**Authors:** Peijun Zhu, Ting Li, Qingqing Li, Yawen Gu, Yuan Shu, Kaibo Hu, Leifeng Chen, Xiaogang Peng, Jie Peng, Liang Hao

**Affiliations:** 1Department of Orthopedics, Second Affiliated Hospital of Nanchang University, Nanchang 330006, China; 2The Second Clinical Medical College, Nanchang University, Nanchang 330006, China; 3Department of General Surgery, Second Affiliated Hospital of Nanchang University, Nanchang 330006, China; 4Jiangxi Province Key Laboratory of Molecular Medicine, The Second Affiliated Hospital of Nanchang University, Nanchang 330006, China

**Keywords:** osteosarcoma, endoplasmic reticulum stress, unfolded protein response, autophagy, oxidative stress, therapy

## Abstract

Osteosarcoma is the most common malignant bone tumor, often occurring in children and adolescents. The etiology of most patients is unclear, and the current conventional treatment methods are chemotherapy, radiotherapy, and surgical resection. However, the sensitivity of osteosarcoma to radiotherapy and chemotherapy is low, and the prognosis is poor. The development of new and useful treatment strategies for improving patient survival is an urgent need. It has been found that endoplasmic reticulum (ER) stress (ERS) affects tumor angiogenesis, invasion, etc. By summarizing the literature related to osteosarcoma and ERS, we found that the unfolded protein response (UPR) pathway activated by ERS has a regulatory role in osteosarcoma proliferation, apoptosis, and chemoresistance. In osteosarcoma, the UPR pathway plays an important role by crosstalk with autophagy, oxidative stress, and other pathways. Overall, this article focuses on the relationship between ERS and osteosarcoma and reviews the potential of drugs or gene targets associated with ERS for the treatment of osteosarcoma.

## 1. Introduction

Osteosarcoma is the most common primary solid malignant tumor of bone. Osteosarcoma is derived from primitive mesenchymal cells, originating from the bone, and is composed of mesenchymal cells that form osteoid and immature bones [1]. Osteosarcoma is also the most common and aggressive bone tumor. The primary locations of osteosarcoma are the proximal tibia and the end of the femur, which are sites that typically form deep tumors [2]. The highest incidence of osteosarcoma is reported in children and adolescents, with a lower incidence in individuals aged more than 60 years [3,4]. The incidence of osteosarcoma has stabilized in the last 40–50 years, and its mortality has decreased; however, the treatment rate for osteosarcoma has not improved substantially in the last two decades. Moreover, owing to the highly locally aggressive and metastatic nature of osteosarcoma, healing in osteosarcoma is considerably poor [5,6].

Treatments and interventions for osteosarcoma have also been explored and researched continuously in recent decades. However, the therapeutic interventions for osteosarcoma are still quite limited. Conventional treatments included radiotherapy, chemotherapy, and surgical resection [7]. The treatments for osteosarcoma currently used in North America and Europe include neoadjuvant multidrug chemotherapy followed by surgical resection for controlling the local tumors at the site of the primary tumor, followed by focal treatment of metastatic sites [8]. However, the key factor that diminishes the effectiveness of osteosarcoma treatment is the resistance of osteosarcoma cells to radiotherapy and chemotherapy. This indicates the urgent need for an efficient and novel treatment for osteosarcoma.

The expression of abnormal genes, such as oncogenes, in cancer cells can promote protein translation and secretion, which consequently increases the pressure on the secretory pathway [9,10] and induces ERS. The high levels of activation and expression of the three major branches of ER markers (IRE1, PERK, and ATF6) and the associated molecular chaperone GRP78 have been reported in numerous human cancers (lymphoma and myeloma) and solid tumors (breast, gastric, lung, liver, esophageal, and colon cancers) in multiple studies [11,12,13]. Additionally, ERS has been shown to play multiple roles in cancer. Sustained moderate ERS induced by the tumor microenvironment, metabolic changes, and oncogenic factors can stimulate cancer cell proliferation and transformation, among other changes, and promote cancer or tumor development, including osteosarcoma [14,15,16,17]. Studies on ERS in cancers or tumors can aid the progress in the treatment of cancer or tumors, and ERS may serve as a novel therapeutic target in cancer or tumors.

## 2. ERS and UPR

The ER, one of the largest organelles in eukaryotic cells, is a three-dimensional network structure that stores Ca2+ in large quantities in its lumen. It is involved in the synthesis, folding, and structural maturation of cellular proteins [18]. Approximately one-third of the proteins in mammalian cells are transported into the ER in an unfolded state, modified and folded by numerous enzymes and chaperones, and eventually transported from the ER in an intact state and transported to various organelles or secreted by the cell [19]. The ability of the ER to correctly fold and induce the post-translational modification of secreted and transmembrane proteins is blocked by various factors, such as genetic and environmental challenges, which cause the accumulation of several misfolded proteins in the ER, inducing ERS [18,20]. ERS occurs in cells with a high level of secretion and is associated with protein misfolding, abnormal protein accumulation, and a host of other challenges, such as altered Ca^2+^ levels, hypoxia, and the presence of reactive oxygen species (ROS) [21].

ERS disrupts ER homeostasis, which activates three quality-control pathways that maintain homeostasis in the ER: the UPR, ER-associated degradation (ERAD), and autophagy. The failure to degrade misfolded proteins activates the UPR, which restores ER homeostasis by suppressing protein synthesis or inducing the expression of genes encoding molecular chaperones and related protein-processing enzymes [20,22]. In addition, components of the UPR activate two major protein degradation pathways: ERAD and autophagy. ERAD can reverse the transfer of misfolded ER proteins into the cytoplasmic matrix, where they are ubiquitinated and degraded by the proteasome. When the accumulation of misfolded proteins exceeds the regulatory capacity of ERAD, autophagy can serve as the secondary response that degrades the misfolded proteins [23,24,25,26].

The UPR is an adaptive cellular response mediated by three sensors: inositol requiring enzyme 1 (IRE1), PKR-like ER kinase (PERK), and activating transcription factor 6 (ATF6). An effective UPR restores ER homeostasis and promotes cell survival, whereas an extreme, insufficient UPR to maintain ER homeostasis leads to cell death [27]. Under physiological conditions, the chaperone glucose regulatory protein 78 kDa (GRP78, also known as BiP or HSPA5) binds to the three UPR sensors and inhibits their activation [27,28], and under ERS, BiP exhibits a greater affinity for misfolded proteins, which resultantly detach from the sensors; this allows the sensors and their downstream signals to be activated [15] (Figure 1).

### 2.1. The Three Branches of UPR

#### 2.1.1. IRE1

IRE1 is a type I ER membrane protein present in animals and yeast and is considered to be the protein with the highest degree of evolutionary conservation [29]. IRE1, PERK, and ATF6 contain three structural domains: an ER luminal domain (LD), a single-pass membrane-spanning domain, and a cytosolic domain [30]. The N-terminal LD interacts with BiP and misfolded proteins within the ER and is used to identify misfolded proteins. The IRE1 C-terminal cytoplasmic region influences cell fate through serine/threonine kinase and endoribonuclease (RNase) domains [28,31]. Under severe ERS, the IRE1 kinase structural domain can trigger cell death by mobilizing c-Jun N-terminal kinase (JNK) and, subsequently, proteins from the pro-apoptotic BCL-2 family [31,32,33,34]. The IRE1 RNase structural domain can induce X-box binding protein 1 (*XBP1*) mRNA splicing and regulate IRE1-dependent decay (RIDD). IRE1 has two distinct isoforms: IRE1α and IRE1β. IRE1α is the major isoform associated with UPR, and both IRE1α and IRE1β perform mRNA splicing and induce RIDD activation, with IREα exhibiting a stronger splicing activity and IRE1β inducing stronger RIDD activation [35,36]. The IRE1 RNase structural domain is activated for the unconventional splicing of *XBP1* mRNA in the cytoplasmic matrix, which produces a modified mRNA isoform (*XBP1s*) encoding the functionally active protein XBP1 (also known as XBP1s). This protein can upregulate the expression of ER chaperones, folding enzymes, and ERAD components in response to ERS [37,38]. The unspliced XBP1 mRNA (*XBP1u*) translates into highly unstable proteins [15]. RIDD can cause the degradation of certain mRNAs, thereby reducing the proteins that enter the ER [39,40].

#### 2.1.2. PERK

PERK is a type I ER membrane protein, and its ER luminal domain is similar in sequence and structure to that of IRE1 LD [41]. When the cytoplasmic kinase structural domain of PERK is activated via trans-autophosphorylation, it inactivates eukaryotic initiation factor 2α (eIF2α). On one hand, eIF2α phosphorylation completely inhibits protein translation, and on the other hand, it facilitates the translation of a specific set of mRNAs [26]. This implies that the activation of PERK can reduce the number of proteins entering the ER as an adaptive response to ERS and, additionally, promote the production of proteins adapted to ERS [42]. eIF2α phosphorylation selectively increases the translation of activating transcription factor 4 (ATF4), which contributes to the expression of genes regulating protein folding, resistance to oxidative stress, and amino acid metabolism, and reduces the accumulation of misfolded proteins in the ER [43]. Under prolonged ERS, ATF4 induces the expression of transcription factor C/EBP homologous protein (CHOP, also known as DDIT3) and upregulates the expression of pro-apoptotic BCL2-related proteins to promote cell death [14].

#### 2.1.3. ATF6

ATF6 is a type II transmembrane protein with two isoforms, ATF6α and AFT6β, similar to IRE1. AFT6α is primarily associated with ERS, and AFT6β primarily plays a regulatory role [36]. Under ERS, ATF6 is transported to the Golgi apparatus, where it is proteolytically hydrolyzed by site 1 and site 2 proteases (S1P and S2P, respectively), and cleaved ATF6 enters the nucleus to regulate the expression of genes encoding the components of XBP1, BiP, and ERAD [44,45,46].

## 3. ERS Signaling in Osteosarcoma

### 3.1. IRE1α-XBP1s Pathway

It has been shown that the IRE1α pathway is associated with apoptosis, differentiation, metastasis, and drug resistance in cancer cells [47,48]. In osteosarcoma, the knockdown of XBP1 leads to tumor growth inhibition [49]. P97, also recognized as valosin-containing protein (VCP) and cytokinesis cyclin 48 (CDC48), belongs to the ATPases associated with multiple cellular activities (AAA+) family [50]. It can affect proteostasis and functions as an important regulator of ERAD [51,52]. The inhibition of P97 in osteosarcoma cells was found to inhibit the ERAD pathway, allowing the accumulation of unfolded proteins in cells, and subsequently activate PERK- and IRE1-related UPR pathways, namely the IRE1α-XBP1s-CHOP pathway and the PERK-eIF2α-ATF4-CHOP pathway, to promote protein degradation and apoptosis [53].

### 3.2. ATF6

Studies have shown that ATF6α activation is an indicator of poor prognosis in osteosarcoma and improves cell survival after chemotherapy [54]. Polo-like kinase 4 (PLK4) belongs to the serine/threonine protein kinase family, and its deficiency inhibits cell proliferation in U2OS osteosarcoma [55]. Under ERS, ATF6 is activated and binds to the PLK4 promoter to recruit C/EBPβ, thereby inhibiting apoptosis in osteosarcoma cells [56]. It was found that upregulation of only pro-apoptotic BH3 family members Noxa and Puma and downregulation of transcription factor E2F1 contributed to ERS-induced cell death under long-term ERS. In osteosarcoma, upregulation of ATF6 and E2F7 during sustained ERS suppressed E2F1 expression, upregulation of E2F7 was dependent on activation of XBP1, and downregulation of E2F1 enhanced Noxa and Puma expression and promoted ER stress-induced apoptosis [57].

### 3.3. GPR78 and GPR94

Both GPR78 and GPR94 are markers of ERS [58]. GRP94 and GRP78 are ER chaperone proteins. Under non-stress conditions, GRP94 and GRP78 maintain ER protein folding environment and protein balance by binding to IRE1, ATF6, and PERK [59]. Under ER stress, GRP78 and GRP94 dissociate from IRE1, ATF6, and PERK, and then bind to unfolded and misfolded proteins to activate UPR. Free IRE1, ATF6, and PERK trigger downstream signal transduction, which increases ER protein folding ability and ER-related protein degradation through transcription of various chaperone-related genes, namely GRP78 and GRP94. Once UPR fails to control the level of unfolded and misfolded proteins, ER activates the apoptotic signal and activates the death factor CHOP to induce apoptosis and cell death [58]. GPR78 and GPR94 are associated with chemotherapy resistance in osteosarcoma. Studies have shown that GRP78 expression is enhanced in osteosarcoma tissues of patients resistant to both doxorubicin and platinum-based drugs, suggesting that GRP78 is involved in ERS-induced drug resistance in osteosarcoma cells. This mechanism of drug resistance may be an adaptive mechanism of cancer cells in response to ERS. By activating UPR, cancer cells upregulate proteins such as GPR78 and GPR94, thus promoting ubiquitination degradation of CHOP protein, reducing the activity of CHOP, and restoring the folding environment and protein balance of ER protein, thus enabling cancer cells to survive [60]. In addition, normal osteoblasts expressed ATF4, which significantly inhibited the occurrence of OS. RET is a receptor tyrosine kinase (RTK) required for normal cell development. The absence of RET upregulates ATF4, which accelerates the turnover of RET proteasomes by over-recruitment of its trans-activated E3 ligase CPL-C, thereby enhancing the chemotherapy effect of BTZ and preventing BTZ resistance in osteosarcoma cells. However, the combination of GPR78 and RET interferes with the interaction between RET and ATF4, down-regulating ATF4 and inhibiting the anti-tumor effect of BTZ, thus leading to chemotherapy resistance [61]. In osteosarcoma, GPR94 does not affect tumor proliferation or migration, but it decreases tumor sensitivity to chemotherapy [62].

### 3.4. Crosstalk of ERS with Other Pathways

#### 3.4.1. Integrated Stress Response (ISR)

The ISR was first identified in 2002 [63,64] and is an adaptive signaling pathway in eukaryotic cells that is activated by a variety of pathological stimuli, including hypoxia, amino acid deprivation, glucose deprivation, and viral infection [65], and the ERS can also activate the ISR [66]. The ISR is mediated by four kinases: PERK, double-stranded RNA-dependent protein kinase (PKR), heme-regulated eIF2α kinase (HRI), and general control non-derepressible 2 (GCN2). Each of these four kinases senses different stimuli, PERK is activated when unfolded or misfolded proteins accumulate in the ER, PKR is activated by binding to double-stranded RNA (dsRNA), HRI is sensitive to heme deficiency, and GCN2 is induced by amino acid deficiency, and they all exert their effects by causing phosphorylation of eIF2α [67]. Under short-term stress, ISR exerts a pro-cell survival effect and restores cellular homeostasis, whereas, under sustained and severe stress, ISR induces apoptosis [68]. PERK-eIF2α-ATF4-CHOP is a common pathway between UPR and ISR, which in osteosarcoma crosstalk with autophagy, oxidative stress, and affects the development of osteosarcoma, as will be described below.

#### 3.4.2. Autophagy

Autophagy is a conserved pathway active in all eukaryotes and it can be triggered by a variety of stress factors, such as nutritional deficiency, hypoxia, oxidative stress, and chemical drugs, and it plays a dual role in cancer cells. On the one hand, it can eliminate oncogenic protein substrates, damaged organelles, and inhibit tumor growth [69]; on the other hand, it can provide nutrition and energy to tumors in a state of hypoxia and nutritional deficiency, and promote tumor cell survival [70] and metastasis [71]. ERS and autophagy were found to be associated with post-chemotherapeutic drug resistance in osteosarcoma, in which PERK plays a key role [72,73,74]. Target of rapamycin (TOR) is a serine/threonine protein kinase that acts as a critical modulator of autophagy, and activation of the mammalian TOR (mTOR) pathway inhibits autophagy [75]. Sestrins (SESN) can regulate autophagy [76]. Recent studies have shown that the cellular expression of Sestrin2, which belongs to the SESN family, increases during chemotherapy for osteosarcoma, which in turn inhibits PERK-eIF2α-CHOP activation, leading to a decrease in CHOP-induced apoptosis. In addition, low CHOP expression results in a decrease in p-mTOR protein levels, and autophagy is subsequently activated, which protects cells by reducing apoptosis [73]. Oh et al. showed that ERS-activated autophagy is an important contributor to radiosensitivity in osteosarcoma [77]. Their study showed that high linear energy transfer (LET) radiation can induce autophagy and promote apoptosis through two ERS pathways, contributing to cellular sensitivity to high levels of LET radiation. Under ERS, PERK is activated, phosphorylates eIF2α, and mediates autophagy through the ATF4-CHOP-Akt-mTOR axis. Moreover, IRE1 is activated, binds to TNFR-associated factor 2 (TRAF2) to form the IRE1-TRAF2-ASK1 complex, and then phosphorylates JNK, thus allowing Bcl-2 to dissociate from Beclin 1 to mediate autophagy (Figure 2).

#### 3.4.3. Oxidative Stress

Oxygen-demanding cells produce ROS during metabolism, and excessive ROS can damage DNA and induce apoptosis [78]. ERS was found to be closely associated with oxidative stress in various diseases [79]. The increase in ROS affects endoplasmic reticulum homeostasis and induces ERS [79,80]. Miao et al. showed that oxidative stress induced by the use of bortezomib in combination with adriamycin for osteosarcoma activated the PERK/eIF2α/ATF4/CHOP axis, thereby inducing apoptosis [81]. Huang et al. demonstrated that drug-laden nanoparticles promoted the generation of ROS in osteosarcoma cells and participated in ERS-induced apoptosis through the JNK/p53/p21 pathway [82]. Unfolded proteins in the ER also activate ROS; the production of approximately a quarter of the ROS content in cells may be related to the formation of disulfide bonds in the ER during oxidative protein folding [83]. One of the mechanisms proposed for the formation of disulfide bonds in the ER that generates ROS is that ER oxidoreductin 1 (ERO1) synergizes with reduced protein disulfide isomerase (PDI) to produce ROS [79]. ERO1 is a producer of H_2_O_2_ in the intracellular UPR pathway and is regulated by CHOP [84,85]. CYT997 (lexibulin) (a microtubule-targeting agent), when used to treat osteosarcoma, damages mitochondria to produce ROS and also induces ERS, which exacerbates oxidative stress through the PERK/ eIF2α/CHOP/ERO1 axis, while ROS produced by the mitochondrial pathway also exacerbates ERS, ERS and oxidative stress enhance each other to promote apoptosis [72] (Figure 2).

#### 3.4.4. PI3K/Akt Pathway

PI3K/Akt is a common and important signaling pathway in cells, which is related to phosphatidylinositol and is also an RTK-mediated derived signaling pathway. AKT is the core of this pathway and numerous molecules and pathways are simultaneously connected downstream, such as VEGF and FOXO [86]. The PI3K/Akt pathway is a key regulator activated during cellular stress [87]. The activation of mTOR can lead to the increased synthesis of various proteins [88]. Tumor cells can escape drug-induced growth inhibition and death through the PI3K/Akt/mTOR pathway [89], whereas the PI3K/Akt/mTOR pathway activates the autophagic pathway, which in turn protects tumor cells [90]. Studies have shown that the inhibition of the PI3K/Akt/mTOR pathway exacerbates ERS in disease cells [91,92]. In addition, NF-κB transcription factors are important regulators of various responses, such as stress response, apoptosis, and differentiation, and are closely linked to other pathways [93], which is more important for the survival and drug resistance of osteosarcoma [94]. The PI3K/Akt/NF-κB signaling pathway is associated with metastasis and invasion in osteosarcoma [95]. Yan M et al. found that when osteosarcoma was treated with cisplatin, the UPR was activated, and both branches of PERK and IRE1 activated NF-κB, thus preventing the cells from acquiring chemoresistance to cisplatin [96]. We also found other evidence that ER stress prevents osteosarcoma cells from acquiring chemoresistance to other drugs besides cisplatin. In osteosarcoma development, PI3K/Akt and its downstream signaling molecules mTOR and NF-κB constitute the PI3K/Akt/mTOR and PI3K/Akt/NF-κB pathways to regulate osteosarcoma through ERS [97,98].

#### 3.4.5. Wnt/β-Catenin Pathway

The Wnt-3a/β-catenin pathway is associated with tumorigenesis and progression [99,100,101]. Some studies have shown that this pathway is associated with ERS [102,103]. However, the specific role of the Wnt-3a/β-catenin pathway in osteosarcoma is still unclear. Several studies have shown that this pathway participates in the regulation of proliferation and apoptosis of osteosarcoma cells; e.g., zinc inhibits the proliferation and promotes the apoptosis of osteosarcoma cells via the activation of the Wnt-3a/β-catenin signaling pathway [104]. MicroRNA-152 suppresses the growth of osteosarcoma cells via the Wnt/β-catenin signaling pathway [105]. However, Yang et al. showed that α-mangostin induced ERS by promoting ROS production, which in turn inhibited the Wnt/β-catenin signaling pathway and activated the caspase-3/8 cascade, causing apoptosis in osteosarcoma cells [106]. At the same time, Jiang et al. demonstrated that histone methyltransferase SETD2 can inhibit the growth of osteosarcoma cells by inhibiting Wnt-3a/β-catenin signaling pathway [107]. We guess that the specific mechanism of Wnt-3a/β-catenin pathway in osteosarcoma is inseparable from this contradiction, which also needs more research to explore and explain.

#### 3.4.6. MicroRNAs

Several microRNAs have been reported to be significantly associated with ERS triggered in osteosarcoma cells, notably miR-1281 and miR-663a [108,109,110]. Recent studies have shown that miR-1281 and miR-663a exert a significant effect on ERS triggered in osteosarcoma cells [111,112]. P53 is a tumor suppressor. USP39 (Ubiquitin precific peptidase 39) plays an important role in osteosarcoma. USP39 knockdown inhibits cell growth and enhances cell apoptosis. In osteosarcoma cells under ER stress, P53 directly binds to the promoter of miR-1281, and down-regulates USP39 in osteosarcoma cells through the p5-Mir-1281-USP39 axis, an ERS response pathway, to inhibit and promote cell apoptosis [111]. ZBTB7A is a member of the POK transcriptional repressor family, which is expressed aberrantly in some cancers and plays an important role in tumorigenesis, including that in osteosarcoma [113,114,115]. ZBTB7A is a target gene of miR-663a. ERS induces miR-663a and protects osteosarcoma from ERS-induced apoptosis by inhibiting the expression of lncRNAGAS5. This change is achieved by Mir-663a-ZBTB7A-lncRNAGAS5 [112].

## 4. Treatment of Osteosarcoma

### 4.1. Current Treatment Methods of Osteosarcoma and Their Limitations

Currently, surgical resection is the most common treatment for osteosarcoma and is combined with radiotherapy for metastatic disease [116]. Osteosarcoma primarily occurs in the distal femur, proximal tibia, and distal femur; thus, even though limb-preserving surgery is widely performed in clinical practice, especially with the development and progress of technology, and postoperative reconstruction methods are more common, early amputation is advised in cases with an inevitable high risk of death from limb-preserving surgery [117]. However, this treatment method causes greater trauma to patients, and osteosarcoma has a high recurrence rate with this treatment, with diverse complications [118].

In addition to surgical treatment, chemotherapy is another common and effective treatment. Combination chemotherapy is commonly used for osteosarcoma, and the chemotherapeutic agents considered to be the most effective are isocyclophosphamide [119], cisplatin [120], adriamycin, and methotrexate [121]. Recently, a greater number of studies have shown that combination therapy (neoadjuvant chemotherapy and adjuvant chemotherapy) can improve the efficacy of radiotherapy without producing extreme side effects while improving healing outcomes [122]. Experimental evidence shows that neoadjuvant chemotherapy and adjuvant therapy exert a significant effect on the healing of osteosarcoma [123]. Link et al. [124] showed that multidrug adjuvant therapy can increase the recurrence-free rate of osteosarcoma from 17% to 66%. Supportive care is a treatment approach for the development of novel drugs that neutralize the toxicity of chemotherapy [125]. Metastatic disease from osteosarcoma is often a significant cause of death, and the most common form of metastasis from osteosarcoma is pulmonary metastasis, which is extremely difficult to treat. Gordon et al. [126] used various mouse models of osteosarcoma to demonstrate that aerosolized chemotherapy caused the regression of lung metastases and improved survival in osteosarcoma mice. Nebulized gemcitabine is an available agent. However, the increasing resistance of osteosarcoma to chemotherapeutic agents has considerably reduced the effectiveness of chemotherapy while causing the recurrence rate of osteosarcoma to increase [127].

According to previous concepts, osteosarcoma is a radiation-resistant tumor. However, recent data indicate a therapeutic role for radiotherapy in patients who have undergone multidrug chemotherapy but are ineligible for complete resection or have tiny tumor lesions remaining after attempted resection; however, the role of this treatment modality remains unclear [1,128].

### 4.2. Potential Osteosarcoma Treatments That Target ERS

The development of new and more effective treatments for osteosarcoma is an interesting pursuit. Since ERS plays a major role in the progression of osteosarcoma, therapeutic strategies that target ERS offer hope in the treatment of osteosarcoma (Table 1).

ZSTK474 is a PI3K inhibitor with an antitumor effect in vitro and in vivo [145]. This drug is involved in therapeutic trials for a variety of tumors, including breast cancer, pancreatic cancer, etc. [146,147,148]. CYT997 has strong cytotoxicity against a variety of cancer cell lines and has anti-vascular effects on the tumor vasculature [149]. Studies have found that CYT997 can promote apoptosis in multiple metastases of head and neck squamous cell carcinoma [150,151]. Melittin has extensive lytic effects in a variety of tumor types and is a clinically common anticancer agent [152] and has been clinically used to treat osteosarcoma. Artocarpin, Kuanoniamine C, Plumbagin, α-Mangostin, and Wogonin are extracted from plants and have a great effect on the treatment of tumors. They have played a huge role in the treatment of other tumors [153,154,155]. Related reports have reported that the above drugs are expected to participate in the treatment of osteosarcoma. More drugs and approaches that could be a potential treatment for osteosarcoma are detailed in Table 1.

#### 4.2.1. IRE1

The use of the P97 inhibitor CB-508 induces the activation of the IRE1α-XBP1s-CHOP pathway and causes apoptosis in cancer cells, and it may serve as an effective inhibitor of osteosarcoma treatment [53]. In addition, the IRE1-ASK1-JNK pathway is typically involved in cell death via ERS; in this pathway, IRE1α activates ASK1 and thus JNK, and JNK phosphorylation inhibits and activates Bcl-2 and Bim, respectively, which eventually leads to cell death. The therapeutic effect of this treatment on osteosarcoma can be investigated, and relevant targeted drug studies have been conducted on this [130,156,157].

#### 4.2.2. PERK

With respect to the use of ERS for the intervention or treatment of osteosarcoma, PERK forms a rather important component [158]. PERK was shown to play a role in both osteosarcoma induction and progression. The main PERK-related pathway is the PERK-eIF2α-ATF4-CHOP-ERO1 axis [159], along with the PERK/mTOR pathway [74]. Under ERS, the PERK-eIF2α-ATF4-CHOP-ERO1 axis affects the translation of proteins that regulate the transcription of multiple genes and induces the initiation of the apoptotic cascade response [159,160]. Changes in the regulation of either class of molecules during the signaling pathway can affect the pathway and prevent the development of osteosarcoma. The phosphorylation of PERK, eIF2α, and ATF4 or the upregulation of CHOP expression enhances signaling. Of these, the phosphorylation of PERK and upregulation of CHOP are the two most common pathways. This pathway has been exploited in numerous studies to develop a novel antitumor drug. PERK-activated autophagy is the primary mechanism by which osteosarcoma cells resist ERS and apoptosis, and the PERK/mTORC1 axis is the key pathway by which PERK activates autophagy, in which PERK inhibits mTORC1 activity, thereby inducing autophagy under stress and promoting the survival of cells [74,161]. Autophagy is currently suggested as the primary mechanism by which osteosarcoma develops resistance to chemotherapy with reduced sensitivity, although the underlying mechanism is unclear [162,163]. However, the PERK/mTORC1 axis provides a potential approach and target for improving chemoresistance in osteosarcoma [74]. In addition, Sestrin2 was shown to be a promising genetic target for increasing chemosensitivity in osteosarcoma [73], and the knockdown of this gene could inhibit cellular autophagy through the PERK-eIF2α-ATF4-CHOP- or PERK/mTORC1 axis, which helps improve chemosensitivity.

#### 4.2.3. PI3K/Akt

The activation of the PI3K/Akt/mTOR axis can interfere with cell growth and survival control and increase competitive growth advantage, and it is an important cause of therapeutic resistance. PI3K inhibitors, Akt inhibitors, and mTOR inhibitors have been developed for tumor therapy [164,165,166], and studies on dual PI3K/mTOR inhibitors, in particular, are currently underway [89,167]. The inhibition of the PI3K/Akt/mTOR pathway can be developed as a novel therapeutic approach for osteosarcoma. In addition, NF-κB inhibition has been shown to significantly promote the sensitivity of osteosarcoma cells to cisplatin, and the targeting of NF-κB is a potential therapeutic strategy [96]. CXCR4 (fusion protein) downregulation induces apoptosis in osteosarcoma cells via the inhibition of the PI3K/Akt/NF-κB pathway, which indicates its suitability as a therapeutic target [97].

#### 4.2.4. Other Therapeutic Targets

ATF6α and GPR94 may serve as prognostic indicators of osteosarcoma [54], whereas GPR94 may serve as a biomarker of the chemotherapeutic response in osteosarcoma [62], suggesting that the targeting of the ATF6α pathway or GPR94 may aid the treatment of osteosarcoma. Furthermore, numerous studies have reported the therapeutic role of Wnt/β-linked protein signaling in cancer/tumor targeting [168] and identified natural Wnt inhibitors [169]. One experiment showed that targeting GRP78 via Wnt/β-linked protein signaling could improve the sensitivity of osteosarcoma cells to photodynamic therapy [106,170]. The mechanism of action of the miR-663aZBTB7A-LncRNAGAS5 pathway in ERS has not been studied extensively, but some experiments have shown that specific drugs can modulate ERS through this pathway and aid the treatment of osteosarcoma [112]. Enhanced GRP78 levels have been reported in various drug studies. Currently, the targeting of GRP78 can stimulate cell death and enhance the sensitivity of osteosarcoma to photodynamic therapy [133,170].

## 5. Future Perspectives

To date, a relatively low number of studies have explored ERS in osteosarcoma, and the pathways by which ERS affects osteosarcoma remain unelucidated. As mentioned previously, autophagy and ERS play important roles during chemotherapy and high-LET radiotherapy, and osteosarcoma progression could be inhibited with a combination therapy targeting these two pathways. As mentioned previously, we also observed that ERS-induced autophagy is increased two-fold in osteosarcoma. Autophagy induced by the inhibition of the Akt-mTOR pathway and activation of the IRE1 pathway promotes apoptosis during high-LET radiation treatment of osteosarcoma [77]. Although during chemotherapy, SESN2 regulates autophagy through the PERK-eIF2α-CHOP signaling pathway to protect cells [73], CYT997 activates the PERK-eIF2α-CHOP axis to induce apoptosis and autophagy in osteosarcoma cells, when the induced autophagy promotes osteosarcoma cell survival [72]. Future studies are still needed to elucidate the level and function of endoplasmic reticulum stress and autophagy in osteosarcoma, which will help enhance our understanding of the crosstalk between endoplasmic reticulum stress and autophagy in osteosarcoma and develop combined therapeutic strategies of autophagy and ERS.

Chondrosarcoma is the second most common primary solid tumor of the bone [171], and fibrosarcoma is a solid tumor with a low cure rate and poor prognosis [172]. ERS is often induced in solid tumors owing to glucose deficiency [66]. According to findings from reviews, GPR78 is more active in solid tumors such as breast tumors [173], fibrosarcoma [174,175], and chondrosarcoma [176]. Potential therapeutic agents targeting ERS in both chondrosarcoma and fibrosarcoma are primarily associated with GPR78. For example, trichodermin [177], 2-amino-3-(2-chlorophenyl)-6-(4-dimethylaminophenyl)benzofuran-4-yl acetate [176], and honokiol [178] can promote GPR78 expression, activate calpain and cysteines pathways, and promote apoptosis, and these may be used for the treatment of chondrosarcoma. Targeting the GPR78 promoter [179] and regulating the expression of GPR78 [180] may also be potential therapeutic strategies for fibrosarcoma. While osteosarcoma is a solid tumor, GPR78 has been found to be associated with it [60]. Thus, we could focus on the development of drugs targeting GPR78 to treat osteosarcoma in the future. Although much remains unknown and uncertain with respect to the role of ERS in osteosarcoma, this is a promising direction for developing therapeutic strategies for osteosarcoma.

## 6. Conclusions

Overall, ERS can exhibit crosstalk with multiple pathways to influence the progression of osteosarcoma, and several compounds associated with ERS have shown potential in the treatment of osteosarcoma; however, these findings are yet to be applied in clinical practice. Chemotherapy resistance in osteosarcoma remains a pressing issue, and we believe that this review provides a comprehensive overview of new perspectives on the treatment of osteosarcoma.

## Figures and Tables

**Figure 1 biomolecules-12-01882-f001:**
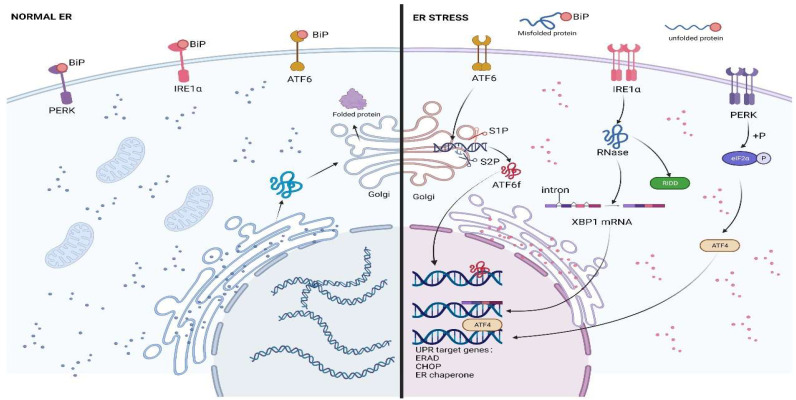
The three branches of the endoplasmic reticulum stress response (ERS) and the unfolded protein response (UPR). When unfolded or misfolded proteins accumulate, the UPR is triggered in the cell to restore endoplasmic reticulum homeostasis. The UPR is transduced by three sensors: IRE1, PERK, and ATF6. Under ERS, Bip is separated from each of the three sensors, which allows them to be activated and subsequently activates downstream transcription factors that promote the expression of UPR target genes (ERAD components, CHOP, and ER chaperones). IRE1 is also involved in the regulated IRE1-dependent decay (RIDD) process to reduce the quantity of proteins entering the ER and alleviate ERS. This figure has been created with https://app.biorender.com (accessed on 20 July 2022).

**Figure 2 biomolecules-12-01882-f002:**
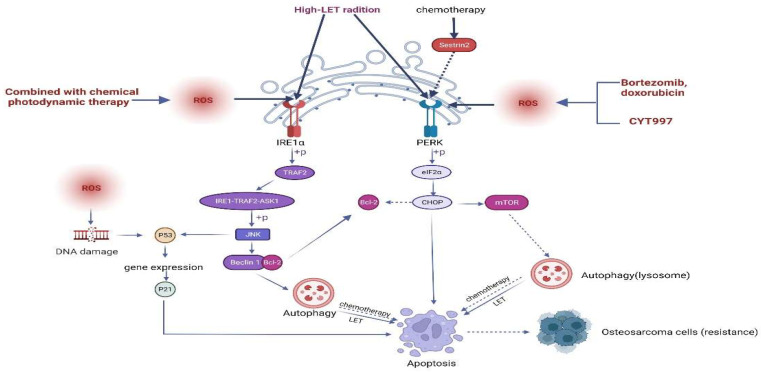
Crosstalk of ERS with autophagy and oxidative stress. The intracellular induction of the Sestrin2/PERK/p-eIF2α/CHOP/mTOR/autophagy axis protects osteosarcoma cells during chemotherapy, leading to the chemoresistance of cells. The intracellular induction of apoptosis via the PERK/p-eIF2α/CHOP/mTOR/autophagy axis and IRE1/TRAF2/ASK1/JNK/autophagy pathway in osteosarcoma cells during high LET radiation treatment induces cellular sensitivity to strong LET radiation. When CYT997 and bortezomib in combination with adriamycin were used to treat osteosarcoma, intracellularly generated oxidative stress induced apoptosis through PERK/eIF2α/CHOP axis regulation. When combined photodynamic therapy was administered, intracellularly triggered oxidative stress in osteosarcoma cells induced apoptosis through JNK/p53/p21 pathway regulation. This figure has been created using https://app.biorender.com (accessed on 20 July 2022).

**Table 1 biomolecules-12-01882-t001:** Drugs or genetic targets associated with ERS for the potential treatment of osteosarcoma.

Medicine	Drug Category	Natural or Chemical Synthesis	Drug Action Target	Mode of Action	Stage of Drug	Reference
ZSTK474	Phosphoinositi-de 3 kinase inhibitor	Chemical synthesis	PI3K/AKT/mTOR	ZSTK474-VSVΔ51 combination therapy	Experimental stage	[98]
CXCR4	Receptor protein	Natural	PI3K/AKT/NF-κβ	Down-regulated expression and inhibition pathway	Experimental stage	[97]
GPR94	Glucose related protein	Natural	GPR94	Upregulated expression and improving chemotherapy sensitivity	Experimental stage	[62]
Sestrin2	Gene target	Natural	Sestrin2/PERK/eIF2α/CHOP	Knocking out and improving chemotherapy sensitivity	Experimental stage	[73]
CYT997	Microtubule targeting agent	Chemical synthesis	PERK/eIF2α/CHOP/ERO1	Activating ROS, inducing autophagy and apoptosis	Experimental stage	[72]
β-Elemonic acid	Active ingredient	Natural	PERK/eIF2α/ATF4/CHOP, Wnt/β-catenin	Direct treatment	Experimental stage	[16]
celastrol	Quinone methylamine triterpenoids	Natural	PERK/eIF2α	Inducing autophagy and apoptosis	Applied treatment	[129]
Surfactant	Cyclic lipopeptide	Natural	IRE1/ASK1/JNK	Abnormal Ca^2+^ Release and strengthening routine treatment	Experimental stage	[130]
E2F1	Transcription factor	Natural	IRE1/Xbp-1	Down, combined with ATF6, inhibition GPR78	Experimental stage	[57]
Melittin	Protein	Natural	IRE1/Xbp-1	Inhibition MG63 cell proliferation	Applied treatment	[131]
Artocarpin	Flavonoid derivatives	Natural	GPR78	Activation of ROS, ER stress and other pathways, inducing apoptosis	Experimental stage	[132]
Kuanoniami-ne C	Nitramine	Natural	GPR78	Degradation GPR78 mRNA, stimulating bortezomib to induce apoptosis	Experimental stage	[133]
α-Mangostin	Enzyme	Natural	Wnt/β-catenin	Inducing apoptosis	Experimental stage	[106]
ZBTB7A	miR663 target gene	Natural	miR-663a/ZBTB7A/LncRNAGAS5	Down regulation, promoting apoptosis	Experimental stage	[112]
Endoplasmic reticulum targeted adriamycin	Adriamycin	Chemical synthesis	C/EBP-β LIP/CHOP/PUMA/caspases 12-7-3	Increasing sensitivity	Experimental stage	[134]
Calcitriol	Alcohol	Natural	Cell cycle	Inhibition of AXT cell proliferation	Approved by FDA	[135]
Plumbagin	Plant isolate	Natural	Apoptotic pathway	Induction of ROS and mitochondrial dysfunction	Experimental stage	[136]
Stim1	Medium factor	Natural	GPR78, CHOP, ATF4	Knocking down, increasing cisplation sensitivity	Experimental stage	[137]
Grphene oxide nanoparticles	MTH1 inhibitor	Chemical synthesis	JNK/p53/p21	Combined photodynamic therapy	Experimental stage	[82]
HA-Lsdox	Conjugated liposome	Chemical synthesis	Pgp, CHOP	Improving the sensitivity of adrianycin treatment	Experimental stage	[138]
Panobinostat	Anticancer drugs	Chemical synthesis	P21, TP53, Bip, CHOP	Inhibiting OS cell survival	Experimental stage	[139]
TIM	Plant isolate	Natural	IRE1, ATF6	Antitumor	Experimental stage	[140]
Wogonin	Flavone	Natural	GPR78	Cleaving GPR78 and promoting apoptosis	Experimental stage	[141]
An-GD2-mAb	Antibody	Chemical synthesis	elF2α, CHOP	Cisplatin and An-GD2 mAb combination therapy	Experimental stage	[142]
GdCl3	Chemical anticancer agent	Chemical synthesis	DNA	Inducing ERS and promoting apoptosis	Experimental stage	[143]
Psoralen	Pseudomonas active ingredient	Natural	ATF6, CHOP	Promoting apoptosis	Experimental stage	[144]

## Data Availability

Data sharing not applicable to this article as no datasets were generated or analyzed during the current study.

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
