# Peer review of "Mechanism and Role of Endoplasmic Reticulum Stress in Osteosarcoma"

_biomolecules, 2022, doi:10.3390/biom12121882_

Round 1
Reviewer 1 Report
Liang Hao and co-workers prepared a literature review concerning the involvement of unfolded protein response (UPR) in osteosarcoma and a potential use of UPR as a target for osteosarcoma treatment.
The manuscript first describes the conditions that cause the ER stress response and the unfolded protein response. The three signalling branches of the UPR (PERK, IRE1 and ATF6) have been reported in some detail. Next the manuscript stated the relationship between ER stress and cancers including the UPR pathways (IRE1 and ATF6) relevant to osteosarcoma. Next, the crosstalk between the ER stress response with autophagy, oxidative stress, the PI3K/AKT and Wnt/beta catenin pathways was exposed. Finally, information is provided on the current clinical management of osteosarcoma and the possibility of using ER stress and some related pathways as targets for osteosarcoma treatment in the future.
The manuscript summarises many original studies, but each concept is often only hinted at, so it is difficult for a reader to extract real information from such telegraphic sentences. This aspect should be improved. Moreover, the information presented in the different sections is often somewhat redundant. By way of example, UPR signalling was described in section 2, then the role of UPR signalling in cancer was emphasised in section 3, and again the UPR in osteosarcoma was outlined (section 4); finally, to suggest that the UPR is a potential target for cancer treatment, the previous concepts were reiterated (section 5.2).
To overcome this problem, we suggest to merge at least sections 4 and 5.2.
Author Response
Thank you for your valuable comments. We have refined some sections with unclear meanings and extended them appropriately. To address the issue of redundant information, we have merged Section 3 with Section 1 and have provided a more comprehensive description of the pathway in what is now Section 3 and added an introduction to the drug in what is now Section 4.2. (L55-64,181-205,209-223,305-307,314-318,321-334,376-386)
Reviewer 2 Report
Summary: Zhu et al. present a rather comprehensive review on endoplasmatic reticulum stress and its role in osteosarcoma presenting ERS and autophagy as therapeutic targets in OS. Overall the manuscript is good and discusses in detail the role of individual components of ERS signaling. However, some sections only present a series of studies without tying the findings together and making a clear point.
Major points:
1. The authors should add a section describing how the ERS response interacts with and overlaps with the integrated stress response (ISR) in OS.
2. In section 4.3, the authors should be clearer on the difference between deficiency and inhibition as they appear to have differing effects. Why?
3. Early-on the authors introduce findings that show a difference between chemo-induced and ERS-induced autophagy; a difference that they resolve toward the end of the manuscript. It would better help the reader to interpret the findings presented if the authors would explain their thoughts on this difference earlier.
4. Section 4.4.2 reads as a list of studies without making a clear point. What is the difference, mechanistically speaking, in oxidative stress by drug treatment that is described versus ROS generation by drug induced ERS and the final results they have on the cell? In what the authors describe, which produces the most oxidative damage?
5. Lines 277-279, refers to cisplatin-induced activation of the UPR inhibiting chemoresistance, but OS often become resistant to cisplatin. The authors should describe the difference between the chemoresistance they refer to and that of chemoresistance that develops to cisplatin.
6. In section 4.4.4, what is the consensus on Wnt/B-catenin, does it promote or inhibit OS. The findings described by the authors appear to be in conflict, and the authors should try to explain this conflict.
7. Section 4.4.5, needs to be restructured to make its point clear. One approach would be to present and discuss each miR separately.
Minor points:
1. Line 103, "iRE1" should be "IRE1".
2. Line 127, add either "family" or "-related" after BCL2.
3. Line 188, "melanoma" should be "osteosarcoma".
4. Line 193, the last part of the sentence should be moved. As is the sentence is somewhat awkward and the author's point is not clear.
5. Line 203, remove "and".
6. Line 241, "eif2a" should be "eIF2a".
7. Line 338, remove the redundant phrase, "resistant to radiotherapy".
Author Response
Reviewer 2:
Q1: Thank you for your valuable comments. We have added a relevant paragraph describing integrated stress response (ISR). While there is little literature on the study of ISR-ERS interactions in osteosarcoma. PERK-eIF2α-ATF4-CHOP is their common pathway, and the role of this pathway in osteosarcoma has been previously described specifically below. (L209-223)
Q2: Thank you for your sincere advice. We re-detailed the role of GPR78 and GPR94 in osteosarcoma. (L181-205)
Q3: Thank you for your valuable comments. We found that ERS is activated during treatment with high linear energy transfer (LET) radiation and during chemotherapeutic drug treatment, and subsequently induced autophagy acts differently. ERS-induced autophagy promotes apoptosis under radiation therapy with high linear energy transfer (LET) radiation, and ERS-induced autophagy protects cells during chemotherapeutic drug treatment. We complement the dual role of autophagy in tumors and the two-sided nature of ERS-induced autophagy in osteosarcoma. (L226-231,453-462)
Q4: Thank you for your valuable comments. Both drug-treated oxidative stress and drug-induced ERS-generated ROS described in the text induce apoptosis via the UPR pathway, and the literature reviewed could not compare the extent of oxidative damage they produce. (L250-251,252-258,264-268)
Q5: Thank you for your sincere advices. We have explained in the text. (L295,297-299)
Q6: Thank you for your valuable comments. The specific role of the Wnt-3a/β-catenin pathway in osteosarcoma is unclear, and based on the current literature, there is no consensus on Wnt-3a/β-catenin in osteosarcoma. (L305-307,314-318)
Q7: Thank you for your valuable suggestions. We introduced and discussed miR-1281 and miR-663a respectively. (L321-334)
Q8: Thank you for your valuable comments. We have corrected “iRE1”. (L112)
Q9: Thank you for your sincere advices. We added “-related” after BCL2. (L136-137)
Q10: Thank you for your valuable comments. We have modified “melanoma”. (L172)
Q11: Thank you for your sincere advices. We changed the part where the meaning was unclear. (L173-179)
Q12: Thank you for your valuable comments. We re-detailed the role of GPR78 and GPR94 in osteosarcoma. (L181-205)
Q13: Thank you for your valuable suggestions. We have corrected “eif2a”. (L266)
Q14: Thank you for your help with the manuscript. We deleted “resistant to radiotherapy” (L365)
Reviewer 3 Report
This review is well thought out and offer a broad overview of possible therapeutic approaches to cure this type of tumor, highly invasive and aggressive. I would invite the authors to add some note/references regarding the use of these drugs, if is just known in literature, about their beneficial effects in this tumor and /or in other pathologies. Further, this aspect underlines the importance of these drugs in the clinical outcome.

Author Response
Reviewer 3:
Thank you for your valuable comments. We have added a description of the drug in the text and added a note in Table 1 about whether the drug is natural or chemical synthesis. (L374-386)
Round 2
Reviewer 1 Report
The revised version of the manuscript has been greatly improved with the addition of specific sentences that help the reader to easily follow the text. Redundant parts have also been reduced by merging different sections. In my opinion, the manuscript deserves publication in IJMS.
Author Response
Dear Editors:
Thank you for your kind letter and your careful work regarding our manuscript. We have revised the manuscript by the reviewer’s comments. And responses to the comments were as follows:
Reviewer 1
Thank you for your patience and careful review of the manuscript, especially the valuable comments you gave us are very helpful to us.
Reviewer 2 Report
The authors have sufficiently addressed my previous comments. There are a number of grammatical errors and typos that need to be addressed.
Author Response
Dear Editors:
Thank you for your kind letter and your careful work regarding our manuscript. We have revised the manuscript by the reviewer’s comments. And responses to the comments were as follows:
Reviewer 2
Thank you for your valuable comments. Sorry for the inconvenience, we have sent the manuscript for editing and proofreading by native English-speaking professionals, and we also checked the manuscript again to fix grammatical errors and spelling errors. (L55,187,201,298,317,326,374,378,385)
